# Photon-photon fusion and tau g-2 measurement in ATLAS

**Agnieszka Ogrodnik⋆ on behalf of the ATLAS collaboration**

AGH University of Science and Technology, Kraków, Poland

⋆ agnieszka.ogrodnik@cern.ch

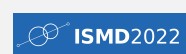 *51st International Symposium on Multiparticle Dynamics (ISMD2022)*
## Abstract

**Relativistic heavy-ion beams at the LHC are accompanied by a large flux of equivalent photons. New measurements of exclusive dilepton production (electron, muon, and tau pairs) performed by the ATLAS experiment are discussed. We present the photon-induced production of tau pairs and constraints on the tau lepton's anomalous magnetic dipole moment. In addition, measurements of photon-induced electron and muon pair production are presented, which provide strong constraints on the nuclear photon flux and its dependence on the impact parameter and photon energy. Forward neutrons are utilised to provide an experimental handle on the impact parameter range sampled in the events.**

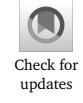
## 1 Introduction

The collisions of ultrarelativistic heavy ions provide a mean to study the strong, weak and electromagnetic (EM) interactions that can occur simultaneously due to multiple nucleon-nucleon interactions. However, the EM interactions become dominant in so-called ultraperipheral collisions (UPC), when two interacting nuclei pass each other at the distance larger than twice the ion radius. The large EM fields associated to ultrarelativistic ions can be considered as coherently produced fluxes of photons, according to equivalent photon approximation [1,2]. Photon-photon interactions occur also in proton-proton collisions, between EM fields of ultrarelativistic protons. However, each photon flux scales quadratically with the ion atomic number, $Z$, what in case of photon-photon beams leads to a $Z^4$ enhancement of the cross-sections for the given process. Therefore, ultraperipheral heavy-ion collisions enable measurements of rare processes, and also searches for new phenomena and particles.

The dilepton photoproduction, $\gamma\gamma \rightarrow \ell^+\ell^-$, where $\ell^\pm$ stands for $e^\pm$, $\mu^\pm$ or $\tau^\pm$ is one of the fundamental processes in UPC. Given the large theoretical uncertainty of photon flux modelling, its precise measurement with exclusive dielectron and dimuon pairs can improve predictions for other photon-induced processes. The results for the $\gamma\gamma \rightarrow \mu^+\mu^-$ process using 2015

lead-lead (Pb+Pb) data and for the $\gamma\gamma \to e^+e^-$ process using 2018 Pb+Pb data collected by the ATLAS experiment [3] at the LHC are discussed in following sections. The exclusive $\tau^+\tau^-$ production is more challenging to measure due to short lifetime of the $\tau$-lepton. However that process was proposed [4–6] to provide the opportunity to constrain the anomalous magnetic moment of the $\tau$-lepton, $a_\tau$, which is sensitive to Beyond Standard Model phenomena. The constraints on the $a_\tau$ set with data collected by the ATLAS detector are also presented.

## 2 Exclusive dimuon production

Exclusive dimuon production is measured by the ATLAS experiment based on 0.48 nb$^{-1}$ of Pb+Pb collision data at $\sqrt{s_{\mathrm{NN}}} = 5.02$ TeV collected in 2015 [7]. The final-state muons have low transverse momenta, $p_{\mathrm{T}}$, and are produced back-to-back in the azimuthal angle. These features are reflected in the definition of the fiducial region of the measurement. Only events with two opposite-charge muons having $p_{\mathrm{T}} > 4$ GeV and $|\eta| < 2.4$ are selected. These requirements are determined by the threshold and acceptance of the muon trigger. Additionally, the dimuon mass, $m_{\mu\mu}$, has to be larger than 10 GeV. Finally, $p_{\mathrm{T}}$ of the dimuon system, $p_{\mathrm{T}}^{\mu\mu}$, has to be below 2 GeV, to ensure a back-to-back topology. After the full event selection, the irreducible background from events with nucleus dissociation remains. These background events occur, when one(both) of the photons is emitted incoherently and the incoming nucleus dissociates (single dissociation). The double dissociation, when both nuclei break up, is also possible.

Distributions of the selected events are compared with Monte Carlo (MC) predictions of the signal process using STARlight [8] generator (for the LO) or STARlight interfaced to Pythia8 [9] generator to account for final-state radiation, FSR. The dissociative background is simulated with LPair [10] generator for $pp$ collisions and normalised in the acoplanarity ($\alpha = 1-|\Delta\phi/\pi|$) distribution using data.

Selected events can be divided into three categories depending on their activity in the forward direction. In the ATLAS detector, this is described with the number of neutrons detected in the Zero-Degree Calorimeters (ZDC): 0n0n class includes events with no neutrons on both sides of the ZDC, Xn0n events have neutrons detected on one side, while XnXn class covers events with neutron signal on both sides of the ZDC. Fractions of dissociative background differ in each ZDC class. Additionally, observed ZDC fractions are affected by the presence of EM pileup and a dedicated correction procedure is applied for this effect.

The integrated fiducial cross-section for the exclusive dimuon production is measured to be: $\sigma = 34.1 \pm 0.3(\text{stat.}) \pm 0.7(\text{syst.})\mu$b, which can be compared with the prediction from STARlight MC generator: 32.1 $\mu$b, and from STARlight+Pythia8: 30.8 $\mu$b. Differential cross-sections are measured in several dimuon variables in the inclusive sample: $m_{\mu\mu}$, absolute dimuon rapidity, $|y_{\mu\mu}|$ and scattering angle in the dimuon rest frame, $|\cos(\theta^*)|$ as presented in Figure 1. In general, a good agreement is found with STARlight, however some deviations are seen, especially at large $|y_{\mu\mu}|$ and small $|\cos(\theta^*)|$.

## 3 Exclusive dielectron production

Exclusive dielectron production is measured by the ATLAS experiment based on 1.72 nb$^{-1}$ of Pb+Pb collision data at $\sqrt{s_{\mathrm{NN}}} = 5.02$ TeV collected in 2018 [11]. The event characteristics are similar to the exclusive dimuon production. In the final state two low-$p_{\mathrm{T}}$ opposite-sign electrons are observed in the back-to-back configuration. The fiducial region is broader than in the dimuon measurement and is defined by requirements on electron $p_{\mathrm{T}}$ to be above 2.5 GeV, $|\eta| < 2.5$, dielectron mass, $m_{ee}$, to be above 5 GeV and $p_{\mathrm{T}}$ of the dielectron system, $p_{\mathrm{T}}^{ee}$, has to

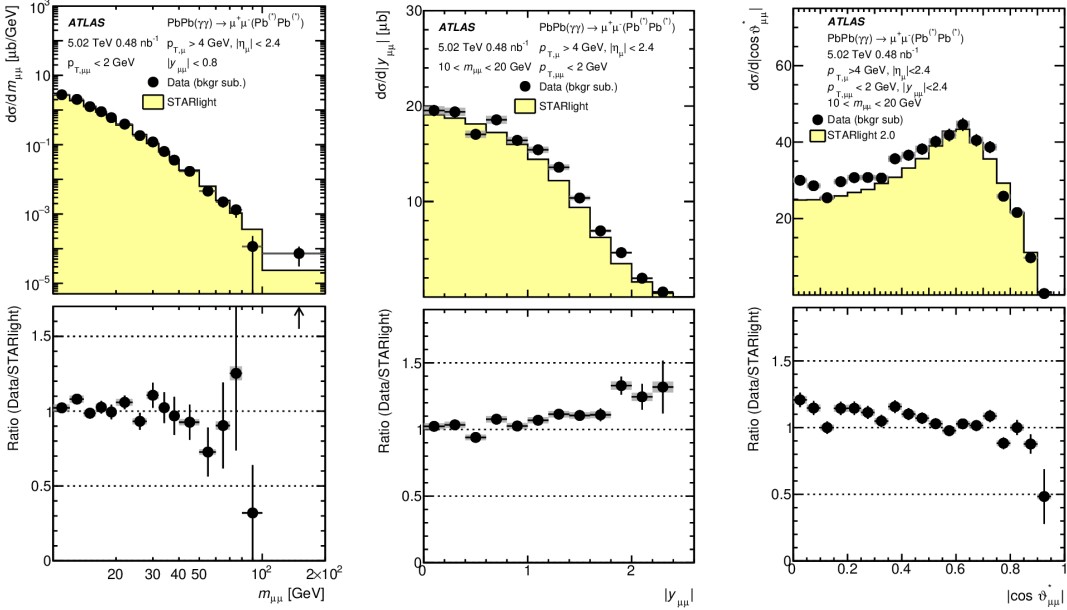

Figure 1: Differential cross-sections for exclusive dimuon production in UPC Pb+Pb collisions as a function of $m_{\mu\mu}$ (left), $|y_{\mu\mu}|$ (middle) and $|\cos(\theta^*)|$ (right) [7]. Data (points) are compared to STARlight predictions (histograms). The statistical uncertainties on the data are shown as vertical bars, while the systematic uncertainties are represented by the shaded bands. The bottom panels present the data-to-simulation ratio.

be below 2 GeV. A background contribution from dissociative production is estimated based on the SuperChic v4 [12] simulation for $pp$ collisions and normalised in the $\alpha$ distribution using data. The background template fitting is performed separately in three ZDC categories, since dissociative background fractions vary in each ZDC class. Additionally, smaller contributions from $\Upsilon$ decays to electrons and exclusive $\tau^+\tau^-$ production are estimated using dedicated MC samples.

The integrated fiducial cross-section for exclusive dielectron production is measured to be: $\sigma = 215 \pm 1(\text{stat.})^{+23}_{-20}(\text{syst.}) \pm 4(\text{lumi.})\mu b$, and is compared to predictions from STARlight: 196.9 $\mu$b, and SuperChic: 235.1 $\mu$b. Differential cross-sections are measured in several variables: $m_{ee}$, absolute dielectron rapidity, $|y_{ee}|$, average electron $p_T$, $\langle p_T \rangle$ and $|\cos(\theta^*)|$ both in the inclusive sample and in the 0n0n class. Results, presented in Figure 2 for differential cross-section as a function of $m_{ee}$ and $|y_{ee}|$ in inclusive sample and as a function of $\langle p_T \rangle$ and $|\cos(\theta^*)|$ in the 0n0n class, are compared to predictions from STARlight and SuperChic MC generators. A data to STARlight ratio is consistent between the inclusive and 0n0n class, and between dimuons and dielectrons. Predictions from STARlight and SuperChic differ mainly in normalisation, however SuperChic describes the rapidity dependence better than STARlight.

## 4 Exclusive $\tau^+\tau^-$ production and constraints on $a_\tau$

The exclusive production of $\tau$-leptons was observed by the ATLAS experiment using 1.44 nb$^{-1}$ of Pb+Pb collision data at $\sqrt{s_{NN}} = 5.02$ TeV collected in 2018 [13]. Due to very low $p_T$ of signal $\tau$-leptons, the identification techniques usually used in ATLAS could not be applied. Instead, signal events are categorised based on the $\tau$-leptons decay modes as $\mu + 1$ track, $\mu + 3$ tracks, and $\mu + e$ categories. Selected events were triggered with a single muon trigger

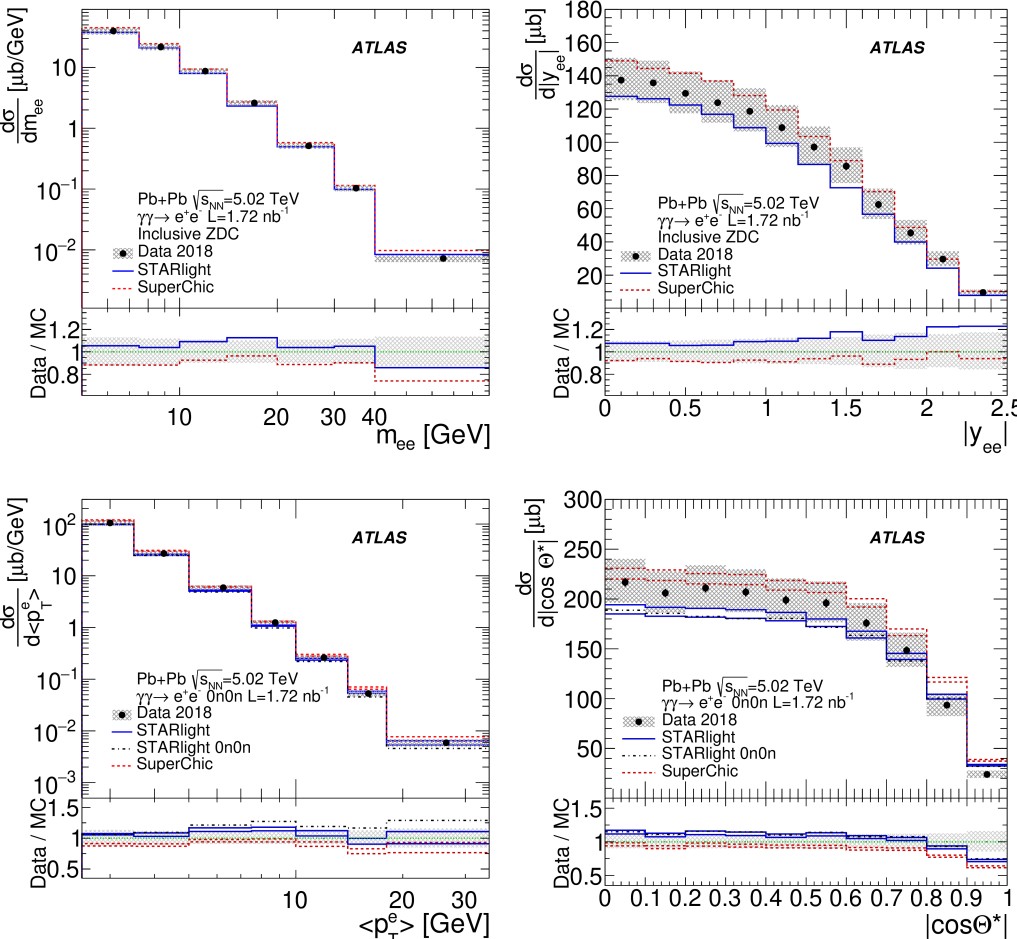

Figure 2: Differential cross-sections for exclusive dielectron production for the inclusive sample: $m_{\mu\mu}$ (top left), $|y_{\mu\mu}|$ (top right) and for the 0n0n class: $\langle p_T \rangle$ (bottom left) and $|\cos(\theta^*)|$ (bottom right) [11]. Data (points) are compared to MC predictions (histograms). The statistical uncertainties on the data are shown as vertical bars, while the systematic uncertainties are represented by the shaded bands. The bottom panels present the data-to-simulation ratio.

requiring muon $p_T$ above 4 GeV. Additionally, the event exclusivity is ensured by imposing a requirement of no forward neutrons detected in the ZDC (0n0n category). For $\mu + 1$ track and $\mu + 3$ tracks categories also events with additional low-$p_T$ tracks and low-$p_T$ clusters are vetoed.

Main background contributions originate from exclusive dimuon production with FSR and diffractive photonuclear interactions. Background from $\gamma\gamma \to \mu\mu(\gamma)$ production is estimated using MC simulation and constrained by a dimuon control region in the data. The $\gamma\gamma \to \tau^+\tau^-$ signal strength and $a_\tau$ value are extracted using a profile likelihood fit using the muon $p_T$ distribution. The fitting procedure is performed simultaneously, combining all signal regions and dimuon control region. The inclusion of the dimuon control region reduces systematic uncertainty from the photon flux. Calculations are based on the same parameterisation as was used in the previous LEP measurements [14]. ATLAS results showed clear, exceeding $5\sigma$, observation of the $\gamma\gamma \to \tau^+\tau^-$ process. The signal strength is consistent with the Standard Model predictions. The best fit value is $a_\tau = -0.041$, with the corresponding 95% CL interval being $(-0.057, 0.024)$. Figure 3 presents the corresponding 95% CL for each of the signal

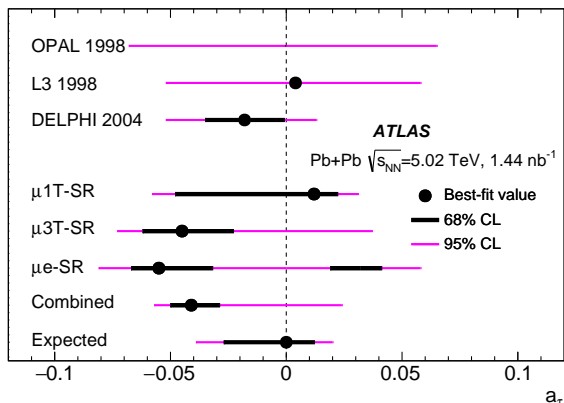

Figure 3: Measurements of $a_\tau$ from fits to individual signal regions, and from the combined fit [13]. These are compared with existing measurements from the OPAL, L3 and DELPHI experiments at LEP. A point denotes the best-fit $a_\tau$ value for each measurement if available, while thick black (thin magenta) lines show 68% CL (95% CL) intervals. The expected interval from the ATLAS combined fit is also shown.

categories including also with the combined value, along with the expected interval and the comparison with the previous measurements of $a_\tau$. The obtained constraints are similar to currently the best result from DELPHI [14]. However, the ATLAS result is largely limited by statistical uncertainty, and could be improved with more UPC data collected in Run 3.

## 5 Conclusions

The exclusive $e^+e^-$ and $\mu^+\mu^-$ productions are measured by the ATLAS experiment at the LHC using ultraperipheral Pb+Pb collision data at $\sqrt{s_{\mathrm{NN}}} = 5.02$ TeV. The measured fiducial cross-sections are compared with theory predictions, showing discrepancies at the level of several percent with STARlight underestimating and SuperChic overestimating measured cross-sections. The results of the differential cross-sections provide a reference for other photon-induced processes and for various theoretical approaches to model the photon fluxes. The $\gamma\gamma \to \tau^+\tau^-$ process has been observed by the ATLAS experiment. The signal strength is consistent with the Standard Model expectations. The constraints on the $\tau$-lepton anomalous magnetic moment are competitive with the previous measurements at LEP and could be improved with a larger data set collected in the future LHC runs.

## Acknowledgements

**Funding information** This work was partially supported by the National Science Centre of Poland under grant numbers 2020/36/T/ST2/00086 and 2020/37/B/ST2/01043, by the program "Excellence initiative - research university" for the AGH University of Science and Technology, and by PL-GRID infrastructure.

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
