# Peer review of "Photon-photon fusion and tau g-2 measurement in ATLAS"

_SciPost Physics Proceedings, doi:SciPost Phys. Proc. 15, 003 (2024)_

## Round 1 · Referee Report · Anonymous (Referee 1) · 2022-11-28

Strengths

  1. A very clear and compact summary of recent ATLAS results in measurements of all three charged-lepton flavours in UPC interactions.

Weaknesses

  1. Some discussion of the meaning, origin, or significance of the model/data deviations and the tau g-2 fit (esp. the bifurcation) would have been nice, but the space was very limited.

  2. Very minor presentational details such as some missing "the" articles and italic mu symbols in units (\textmu helps). But none obstructing comprehension.

Report

A very good summary of the experimental activity in this area. Very suitable for publication in these proceedings. I recommend immediate publication and if the author feels they want to optionally resubmit with any additions in response to the comments, I am sure that can be accommodated.

Requested changes

None

---

## Editorial Decision

published